# Investigating the Role of Ethical Self-Identity and Its Effect on Consumption Values and Intentions to Adopt Green Vehicles among Generation Z

**Muhammad Yaseen Bhutto [1],*, Mussadiq Ali Khan [2], Myriam Ertz [3]** and **Haowei Sun [4]**

1   Business School, Shandong Jianzhu University, Jinan 250101, China
2   Faculty of Economics and Business, Universiti Malaysia Sarawak, Kota Samarahan 94300, Malaysia; 18010098@siswa.unimas.my
3   Labo NFC, Department of Economics and Administrative Sciences, University of Quebec in Chicoutimi, Saguenay, QC G7H 2B1, Canada; myriam_ertz@uqac.ca
4   School of Art, Shandong Jianzhu University, Jinan 250101, China; sunhaowei@sdjzu.edu.cn
*   Correspondence: yaseenbhutto78@yahoo.com or bhutto.yaseen@hotmail.com

**Abstract:** Consumption values and consumer identities have been widely explored as needed in the literature on sustainability, but they are mainly treated as separate predictors of sustainable behavior. Researchers are calling for further analysis of these variables' combined impacts to investigate sustainable behavior. This research integrates consumption values theory (functional, social, conditional, epistemic, and emotional) and ethical self-identity to explore the intention to adopt green vehicles among Generation Z. The data in this paper were collected from 319 respondents through a structured questionnaire in two universities in Islamabad, Pakistan. We used the PLS-SEM approach to analyze the results; we found that functional value (quality), social value, conditional value, and emotional value significantly influenced the intention to adopt green vehicles among Generation Z. The study further demonstrated that ethical self-identity significantly mediates the relationship between consumption values and the intention to adopt green vehicles among Generation Z. Our findings indicate that ethical self-identity inferences are favorable when promoting green vehicles among Generation Z. Therefore, the results of this study provide novel understandings for marketers and policymakers in Pakistan to emphasize the improvement of consumer values and ethical self-identity, which will eventually contribute to the adoption of green vehicles. In addition, automakers should promote green ideas, to encourage Generation Z to replace their existing vehicles with alternative green options in Pakistan.

**Keywords:** sustainable consumption; consumption values; ethical self-identity; intention to adopt green vehicles

## 1. Introduction

The Industrial Revolution and rapid commercial growth, particularly over the past two centuries, have steered society toward excessive consumption and damage to natural resources. As an outcome, today's world faces significant ecological challenges, such as water and air pollution, soil degradation, climate change, and deforestation [1]. One of the critical causes of environmental pollution is increased car ownership and usage [2,3]. According to the International Energy Agency (IEA), there are presently around 1 billion vehicles globally that consume 60 million barrels of oil per day (approximately 70% of overall oil production); private vehicles use an average of approximately 36 million barrels of oil per day [4]. In addition, transport accounts for 23% of global carbon dioxide ($CO_2$) emissions, an essential component of greenhouse gases, and contributes to global warming [5]. Therefore, one way to reduce ecological harm is by adopting green vehicles [2,6]. Current research has deliberated various aspects of consumer behavior pertaining to green

vehicles [7]. For example, multiple studies report that the features of environmentally friendly vehicles (e.g., less pollution, resource conservation, and energy efficiency) can stimulate consumer sentiments to protect the environment and will affect their consumption behavior [2].

Similarly, situational factors, such as ecological problems or subsidies on green vehicles, also inspire people to choose green vehicles [8]. Social surroundings and the views of their peers also inspire green purchase decisions in consumers [9]. In addition, consumer environmental knowledge and awareness have also been reported as significantly impacting the consumer's decision-making process for green vehicles [10]. Finally, consumers' desire to look for novelty and to gain significant insight into the product's usefulness in terms of quality and price can also influence their choice to purchase a green vehicle [11]. In summary, the literature suggested that the main drivers behind consumers' purchasing decisions for green vehicles are the consumption values that consumers experience when using the product.

As Pakistan is the seventh most greatly affected country due to climate change, it needs to consider moving toward ecological energy solutions [12]. The country faces more deaths due to air pollution, emphasizing the highly severe ecological circumstances for public living [13]. According to the latest broadcasting reports [14], the Pakistani government is keen on e-mobility and aims to convert 90% of the vehicle fleet to electric vehicles by 2040 [15]. Green adoption helps to mitigate the high dependence on fossil fuels and reduce $CO_2$ emissions [4,16]. Consequently, understanding consumer behavior toward the adoption of green vehicles is essential for decision-makers, manufacturers, and marketers [17]. Green vehicle adoption has been widely investigated in both developed and emerging economies [2,7,8,10,18–21]. However, there are fewer studies on green vehicle adoption in developing economies (predominantly Asian counterparts), such as Pakistan [22–24].

To understand how to encourage consumers toward sustainable consumption, earlier studies have examined the role of consumer identity in determining sustainable consumption behavior [25,26]. In this aspect, some current studies claim that consumer self-identity can help forecast an extensive range of eco-friendly inclinations, intentions, and behaviors [27,28]. In light of this, some recent studies claim that consumers demonstrate particular kinds of self-identities that may be useful for forecasting a comprehensive range of pro-environmental behavior, such as green self-identity [21], organic food identity [29], environmental self-identity [26], and ethical self-identity [30]. Recent studies have examined the role of ethical self-identity in adopting sustainable consumption [30,31]. According to Michaelidou and Hassan [30], ethical self-identity is a significant predictor of consumer attitude and purchase intention of organic products. In addition, Sciarelli et al. [32] also mentioned that ethical self-identity enhances consumer attitudes toward ethical consumption. Furthermore, Talwar et al. [33] suggested that self-identity could be an important determinant of behavior, arguing that future research should explore the role of ethical self-identity. Our study differs from previous studies in that we examine whether ethical self-identity mediates the connection between consumer values and the intention to adopt green vehicles. Therefore, in contrast to previous research (e.g., [32]), we posit that consumer ethical self-identity might significantly mediate the relationship between consumption values and consumer green vehicle adoption behavior.

Until now, the literature has focused on various aspects of green vehicles, emphasizing the environmental (e.g., reduced $CO_2$ emissions) and technological (e.g., mileage) aspects [8,20,34–37]. For instance, researchers have mentioned generational impacts and differences in sustainable consumption [38–41]. However, the literature suggests that not all generations are the same, and marketers cannot approach them all in the same way [42]. For example, Generation Y members (born 1982–1994) are generally highly educated, adventurous, and talented at performing multiple challenging tasks [43]. The subsequent emerging consumer generation is Generation Z (1995–2010) [44]. This cohort differs from previous generations in their growing desire for sustainable products [45]. Affiliates of this generational cohort are well-educated consumers who have a better understanding of sustainability issues and sustainable products [46]. While the studies above provide a good overview of the adult population, the younger generations (Generation Z) are notoriously understudied. Given that they represent future buyers of green vehicles and will thus play a significant role in transforming the transport sector, this is a significant shortcoming of the extant literature. Therefore, this study aims to shed additional light on Generation Z's intentions toward green vehicles, an important generation that has received very little attention from researchers, by answering the following research questions.

- How do consumer values influence Generation Z's intention to adopt green vehicles?
- What is the impact of incorporating ethical self-identity as a mediator in the relationship between consumer consumption values and Generation Z's intention to adopt green vehicles?

Understanding Generation Z's intention to adopt green vehicles may illuminate past topical research on adopting green vehicles in young and developing economies. Besides, the results may facilitate managers' and policymakers' efforts towards creating sustainable strategies.

## 2. Literature Review

### 2.1. Theoretical Framework

The impact of values on environmentally friendly decisions and behavior was fully supported [47,48]. Studies have highlighted the multi-dimensional conceptualization of consumer value as an important element in forecasting consumer decisions [49]. Sheth et al. [50] developed a theory of consumption values. Based on this theory, behavior is determined by five values. These values are classified into functional, emotional, social, epistemic, and conditional values. According to the theory of consumption values, perceived value determines the consumer's purchasing decisions and brand choice [50].

The literature recommends that individuals' identities mediate the relationship between their motives and behaviors [26]. This could be expressed from a hierarchical motivational-identity-behavioral viewpoint [51]. Ajzen and Fishbein [52] demonstrate that identity can interfere in the relationship between consumer psychographics and behavior [53]. Johe and Bhullar [54] discussed the theory that consumers select products that match their identity. Previous studies have studied the impact of consumption values on sustainable consumer behavior [26]. In addition, the preceding studies have reviewed the influence of self-identity on sustainable consumer behavior [53,55]. In addition, recent studies have concentrated on the relationship between consumer motivation, self-identity, and sustainable behavior [21,56]. This study explores the theory that consumption values (functional, social, conditional, epistemic, and emotional) indirectly influence consumer intention to adopt green vehicles through ethical self-identity. The proposed conceptual model is shown in Figure 1.

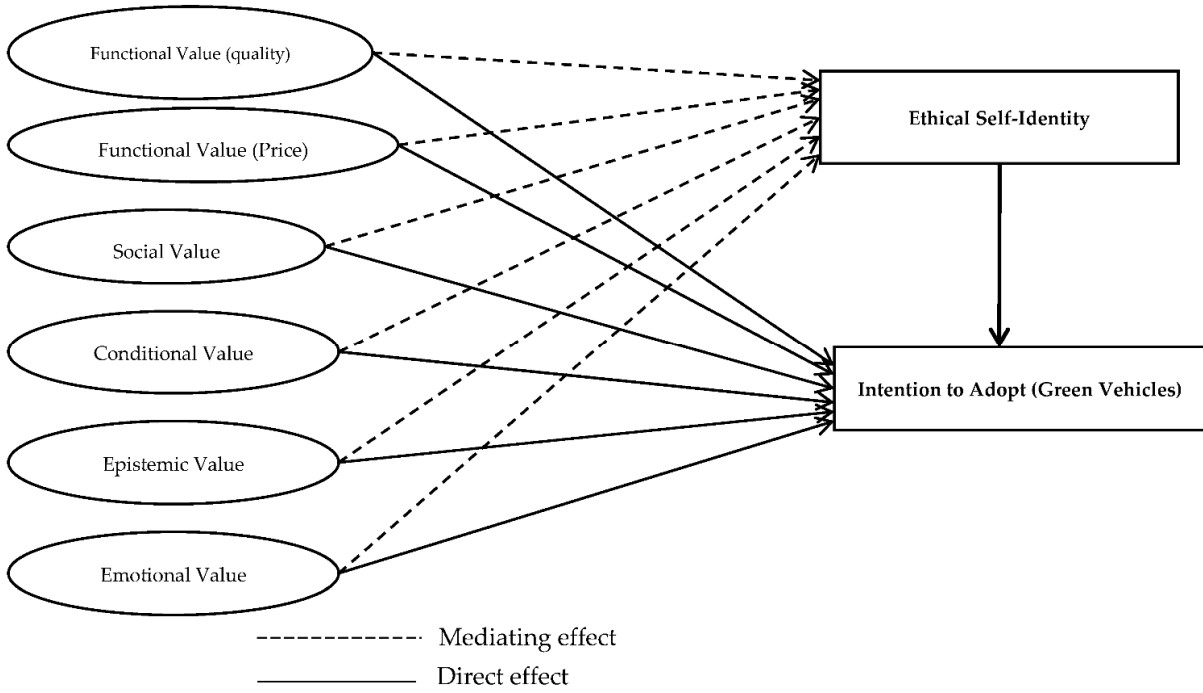

**Figure 1.** Conceptual research model.

*2.2. Hypothesis Development*

2.2.1. Functional Value and Intention

Functional value is a significant driver of consumer choice for sustainable products [57]. It expresses the way consumers recognize a product's performance in terms of price, dependability, durability, and reliability [50]. Based on these features, functional value can be allocated into two dimensions: quality and price [26]. Consumers evaluate the price and quality of a green vehicle when making a buying choice [58]. A consumer may select a green vehicle according to specific attributes, such as environmental benefits, driving comfort, reliability, convenience or operability, charging time, and driving range [59]. In addition, consumers also evaluate the performance of green vehicles in terms of their ability to deliver monetary value. If the product's benefits are considered to justify the price paid for green vehicles, it results in a willingness to pay a higher price for green vehicles [58]. Therefore, quality and price are key factors in consumers' decisions to choose green vehicles. Based on this discussion, the authors have developed the following hypothesis:

**Hypothesis 1 (H1). a**. *Functional value (quality) positively affects consumers' intention to adopt green vehicles.* **b**. *Functional value (price) positively affects consumers' intention to adopt green vehicles.*

2.2.2. Social Value and Intention

Social value generally expresses the degree of approval or disapproval of certain behaviors by people, society, or other important persons [60]. In the preceding literature, social value incorporates social image, social self-image, identification, expressions of personality, and the pursuit of belonging to a particular social class [61]. Kumar and Ghodeswar [62] claimed that consumers would be happy to have their environmental contributions recognized or praised by others. Previously, scholars have found that consumers' purchase selections were influenced by social factors, such as friends and family [63]. In the case of green vehicles, social value is a perceived net benefit gained from using them, based on the point of view of social pressure or the acquisition of status. Social pressure encourages consumers' adoption intention toward green vehicles [2]. Recently, Han et al. [58] found

that subjective value does not significantly influence consumers' intention to adopt green vehicles. The lack of a relationship suggests that other people's opinions may not easily influence consumers. Unconvincing results from the studies available suggest that the relationship between social value and behavioral intention needs more research dissemination. Thus, a second hypothesis is suggested in this study:

**Hypothesis 2 (H2).** *Social value positively affects consumers' intention to adopt green vehicles.*

### 2.2.3. Conditional Value and Intention

Conditional value represents the benefits that choosing an alternative will bring for decision-makers, based on particular situations and circumstances [50]. It occurs when the use of a product is strongly related to specific situations [50].These particular situations may include support or discount for environmentally friendly vehicles, pollution levels, the range of driving distance, charging time, and convenience [58]. Consumer research has recognized that changes in situational variables can significantly impact behavioral intentions [64]. In addition, a conditional value significantly influences consumer pro-environmental behavior [26,65]. Past studies have mentioned tax refunds, government rules and regulations, and government subsidies as influencing green vehicle adoption [66,67]. Therefore, the authors propose the following hypothesis:

**Hypothesis 3 (H3).** *Conditional value positively affects consumers' intention to adopt green vehicles.*

### 2.2.4. Epistemic Value and Intention

Epistemic value is "the perceived utility acquired from an alternative's capacity to arouse feelings or affective states" [50]. Pihlström and Brush [68] describe epistemic value as novelty value and the benefit of learning new ways of doing things. Epistemic value increases when a person who uses or experiences a new product or service is bored with an existing one; they are looking for something else or want to satiate an interest in something new [69]. The consumers' desire to learn information about product innovation, relevance, and product characteristics can significantly affect their behavioral intentions [26,70]. A significant positive impact of epistemic value has been reported on consumers' purchasing behavior for green products [49]. Consequently, the authors propose the following hypothesis.

**Hypothesis 4 (H4).** *Epistemic value positively affects consumers' intention to adopt green vehicles.*

### 2.2.5. Emotional Value and Intention

Emotional value stems from the product's capability to stimulate consumers' feelings or affective states [50]. Emotional values also have a significant impact, particularly on consumers' adaptation to new technologies [71]. Furthermore, experiences and feelings related to past product consumption help predict the future consumption outcomes of individuals with substantial emotional value [72]. Previous studies have reported a substantial positive effect of emotional value on consumers 'purchasing behavior regarding green products [26,48,73]. Thus, the authors propose the following hypothesis.

**Hypothesis 5 (H5).** *Emotional value positively affects consumers' intention to adopt green vehicles.*

### 2.2.6. Ethical Self Identity

Identity research is of great importance in shaping pro-environmental behavior [73]. Several studies have recommended the significance of self-identity in predicting ecological preferences, intention, and behavior [25,53]. Self-identity is the composition of the collected roles that an individual fulfills, which encourages continuous action to accept self-perception [74]. Hence, ethical self-identity refers to the degree to which ethical motives

guide a consumer in making consumption choices [75]. Furthermore, ethical consumption includes "conscious and deliberate" consumption decisions, based on individual beliefs and moral concepts [76]. Pelsmacker et al. [73] argue that consumers engage in ethical consumption to protect the environment and society. Ethical self-identity and motives strongly influence consumer behavioral intention [30,77].

In the present study, the authors suggest that consumption values positively correlate with consumers' intentions to adopt green vehicles. Furthermore, the consumers' ethical identity significantly mediates this positive relationship. The values that consumers derive from green vehicles motivate them to consume the product and fulfill their moral self-identity to be environmentally friendly individuals. For example, the functional value derived from green vehicles provides functional benefits, such as monetary benefits and superior quality. In addition, it affects an individual's ethical self-identity by reinforcing their moral perception that utilizing green vehicles helps protect the environment and society while reducing environmental pollution.

Similarly, on the one hand, the social value associated with green vehicles elevates a person's societal image and status. On the other hand, it satisfies ethical self-identity via the knowledge that one is making the morally proper choice in the best interests of the environment and society. Likewise, the conditional value attached to a green vehicle provides consumers with situational benefits in terms of subsidies and rebates, while simultaneously influencing their ethical self-identity to do what is morally right in deteriorating ecological conditions. Finally, the epistemic value resulting from the consumption of green vehicles meets the knowledge needs of the product and provides information about the moral contribution of individuals to society. Building on a similar logic, the recent literature recognizes that consumer self-identity mediates the relationship between consumer motivations/values and behavioral intention for sustainable products [26,29].

**Hypothesis 6 (H6).** *Ethical self-identity mediates the effect of: (a) functional value (quality) (b) functional value (price) (c) social value, (d) conditional value, (e) epistemic value, and (f) emotional value on the intention to adopt green vehicles.*

### 3. Research Methodology

The target population for this study is Generation Z (people born between 1995 and 2010) [44]. This ensures that respondents had some information and interest in the product category, which improved the ability of the self-reporting method to predict behavior. Furthermore, the study targeted educated consumers since they could efficiently respond to the survey. In addition, the concepts of green vehicles and their consumption in young consumers are increasingly accepted. A total of 350 surveys were distributed to two different universities in Islamabad. After the content screening, 319 were eventually retained, which constitutes a response rate of 87%, excluding incomplete responses and extreme outliers. The respondents' participation was entirely voluntary and no incentive or reward was provided before or after the interview. The convenient sampling technique was used because it is suitable for information-gathering when obtaining a complete sampling frame is challenging. Furthermore, this kind of sampling is appropriate because it allows the generalization of the outcome. According to Rahi [78], the minimum sample size should range between 200 and 500 responses. Therefore, 319 valid questionnaires were adequate for evaluating the data. The demographic details of the respondents can be found in Table 1.

**Table 1.** Respondent profiles.

| Category | Frequency | Percentage |
|---|---|---|
| Gender | | |
| Male | 240 | 75% |
| Female | 79 | 25% |
| Age | | |
| Under 18 years | 51 | 16% |
| 18–23 years | 268 | 84% |
| Education | | |
| Undergraduate | 38 | 12% |
| Masters and higher | 281 | 88% |
| Household Income | | |
| Less than 50,000 RS | 56 | 17% |
| 50,001–100,000 RS | 182 | 57% |
| ≥100,001 | 81 | 26% |

*3.1. Measurements*

The data for this research has been collected through a questionnaire. The items to measure functional value (quality), epistemic value, and conditional value were modified from Zailani et al. [79]; three items for evaluating price were adapted from Haroon et al. [26] and three items for social value were adapted from Huang et al. [80]. In addition, three items were adapted from Brich et al. [81] to measure ethical self-identity. Finally, four items were adapted from Mamun et al. [2] to measure the respondents' intention to adopt green vehicles. All items were measured on a scale ranging from "strongly disagree" (1) to "strongly agree" (5), a 5-point Likert scale. All items used in the questionnaire are listed in Appendix A.

*3.2. Common Method Bias*

Data were collected from the same respondent for both predictor and dependent variables through the same instrument/questionnaire, using the same method (survey); therefore, there may be a problem known as common method bias [82]. Harman's one-factor test was performed, using SPSS 16.0, to check for common method biases. Harman's one-factor test is a non-rotated exploratory factor analysis performed on the instrument or questionnaire. In this test, if a single factor cannot explain most (0.50) of the variance, there is no common method bias problem. In this study, the single factor could only explain 38% of the variance of the whole instrument; therefore, there is no common method bias.

## 4. Data Analysis

This section describes the partial least squares structural equation modeling (PLS-SEM) conducted to analyze the conceptual model, using SmartPLS 3.0. The analysis was carried out in two stages. First, the measurement model was evaluated with SmartPLS to assure the validity and reliability of the abovementioned measurement scales. Second, the structural model was assessed to test the hypothetical relationships. according to the two-step analysis process for SmartPLS.

*4.1. Measurement Model*

Partial least squares structural equation modeling (PLS-SEM) was used to assess this study's measurement and structural models. PLS-SEM is a method that can be useful for small sample sizes; it is appropriate for theory development and not required data normality [83]. We have followed two approaches in this paper. First, we considered

the reliability and convergent validity of the measurement model. We then assessed the structural model. According to Hair et al. [84], the data are reliable and internally consistent when Cronbach's alpha values exceed 0.70. The measurement model of this study shows that Cronbach's alpha values for all constructs are above the threshold of 0.70. Composite reliability (CR) values for all constructs exceed 0.70, endorsing the belief that the data are internally consistent [84]. To check convergent validity, the CR values need to be better than 0.70, and the average variance extracted (AVE) values must be greater than 0.50 [85]. Convergent validity determines that the values of CR and AVE are more significant than the suggested cut-offs, as indicated in Table 2 and Figure 2.

**Table 2.** Factor loadings, composite reliability, and average variance extracted (AVE).

| Constructs | Items | Loadings | Cronbach's Alpha | CR | AVE |
|---|---|---|---|---|---|
| Functional Value (Quality) | | | 0.843 | 0.905 | 0.761 |
| | FQ1 | 0.881 | | | |
| | FQ2 | 0.847 | | | |
| | FQ3 | 0.889 | | | |
| Functional Value (Price) | | | 0.731 | 0.872 | 0.775 |
| | FP1 | 0.807 | | | |
| | FP2 | 0.948 | | | |
| | FP3 | 0.889 | | | |
| Social Value | | | 0.732 | 0.848 | 0.654 |
| | SV1 | 0.665 | | | |
| | SV2 | 0.876 | | | |
| | SV3 | 0.886 | | | |
| Conditional Value | | | 0.771 | 0.871 | 0.699 |
| | CV1 | 0.608 | | | |
| | CV2 | 0.947 | | | |
| | CV3 | 0.912 | | | |
| Epistemic Value | | | 0.827 | 0.896 | 0.742 |
| | EP1 | 0.880 | | | |
| | EP2 | 0.830 | | | |
| | EP3 | 0.873 | | | |
| Emotional Value | | | 0.721 | 0.829 | 0.619 |
| | EV1 | 0.780 | | | |
| | EV2 | 0.714 | | | |
| | EV3 | 0.860 | | | |
| Ethical Self-Identity | | | 0.813 | 0.889 | 0.727 |
| | ESI1 | 0.882 | | | |
| | ESI2 | 0.841 | | | |
| | ESI3 | 0.835 | | | |
| Behavioral Intention | | | 0.802 | 0.871 | 0.631 |
| | BI1 | 0.749 | | | |
| | BI2 | 0.858 | | | |
| | BI3 | 0.782 | | | |
| | BI4 | 0.683 | | | |

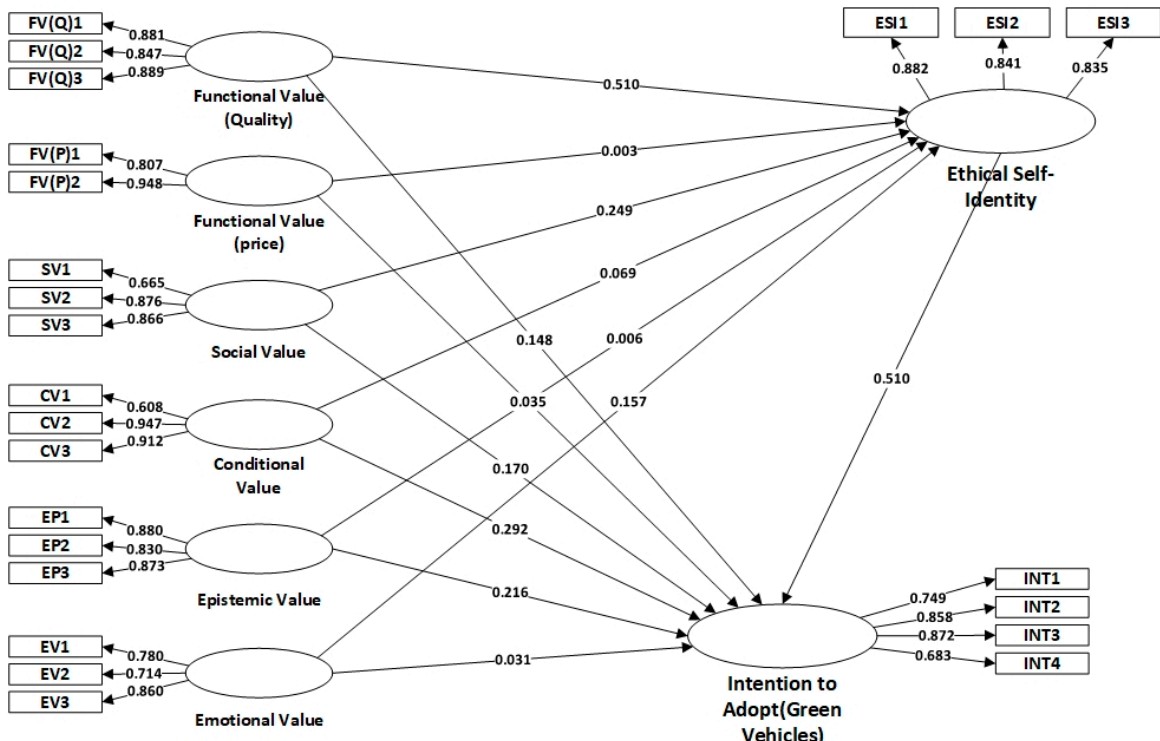

**Figure 2.** Measurement model.

Discriminant validity refers to the degree to which a construct is separate from other constructs [84]. In this study, we have used the Fornell and Larcker criteria. According to Fornell and Larker (1981), discriminant validity is confirmed when the square root values of all AVEs are higher than the corresponding correlation values. Table 3 reveals that the square root values of the AVEs are better than the corresponding correlation values that prove discriminant validity.

**Table 3.** Discriminant validity.

|  | CV | EV | EP | ESI | FV(P) | FV(Q) | INT | SV |
|---|---|---|---|---|---|---|---|---|
| Conditional Value | 0.836 | | | | | | | |
| Emotional Value | 0.482 | 0.787 | | | | | | |
| Epistemic Value | 0.696 | 0.508 | 0.861 | | | | | |
| Ethical Self-Identity | 0.620 | 0.576 | 0.571 | 0.853 | | | | |
| Functional Value (Price) | 0.443 | 0.222 | 0.432 | 0.425 | 0.880 | | | |
| Functional Value (Quality) | 0.633 | 0.544 | 0.590 | 0.818 | 0.430 | 0.872 | | |
| Intention to Adopt (Green Vehicles) | 0.781 | 0.526 | 0.732 | 0.790 | 0.456 | 0.776 | 0.794 | |
| Social Value | 0.626 | 0.446 | 0.590 | 0.733 | 0.562 | 0.731 | 0.755 | 0.808 |

Note: CV = condition value; EV = emotional value; EP = epistemic value; ESI = ethical self-identity; FV (P) = functional value (price); FV (Q) = functional value (quality); INT = intention to adopt green vehicles; SV = social value.

### 4.2. Structural Model Assessment

For the structural model, beta coefficients, t-statistics, and measures of R2 for the endogenous constructs are considered [86]. The bootstrapping method, based on 5000 samples, was used to measure the path coefficients and their relative significance. Using these standard calculations, we measure the impact size (f2) for each structural pathway as suggested by Hair et al. [87]. The R2 values for ethical identity and the intention to adopt green vehicles are 0.730 and 0.804, respectively.

### 4.3. Direct Relationships

Table 4 shows the beta coefficient values, significance values, and effect sizes (f2) for each structural pathway. The results disclose functional value (quality) ($\beta$ = 0.148; *t*-value = 2.256; *p*-value $\leq$ 0.01), social value ($\beta$ = 0.170; *t*-value = 2.622; *p*-value $\leq$ 0.01), conditional value ($\beta$ = 0.292; *t*-value = 5.203; *p*-value $\leq$ 0.01) and emotional value ($\beta$ = 0.216; *t*-value = 4.021; *p*-value $\leq$ 0.01) have a significant positive influence on the intention to adopt green vehicles. Among these findings, conditional values have a strong influence on the intention to adopt green vehicles, followed by emotional value. Therefore, hypotheses H1(a), H3, H4 and H6 are supported, while functional value (price) ($\beta$ = 0.035; *t*-value = 0.905; *p*-value $\geq$ 0.05), epistemic value ($\beta$ = 0.009; *t*-value = 0.226; *p*-value $\geq$ 0.05) and epistemic value ($\beta$ = 0.031; *t*-value = 0.667; *p*-value $\geq$ 0.05) have a non-significant influence on the intention to adopt green vehicles. Therefore, the findings do not support hypotheses H1(b) and H5. Furthermore, the authors rated the measurement of effect strengths, such as 0.02, 0.15, and 0.35 for small, medium, and large effects, as proposed by Cohen [88]. Functional value (quality), social value and emotional value exceeded the threshold criteria of 0.02, indicating a small to medium effect, and conditional value exceeded the threshold criteria of 0.15, indicating a small to medium effect, as shown in Table 4. Conversely, functional value (price) and epistemic value revealed no applicable significance.

**Table 4.** Structural relationship of sustainability values.

| Relationships | Beta Value | *t*-Value | *p* Values | F Values | Results |
|---|---|---|---|---|---|
| H1(a): FV(Q) $\rightarrow$ INT | 0.148 | 2.526 | 0.012 | 0.032 | Accepted |
| H1(b): FV(P) $\rightarrow$ INT | 0.035 | 0.905 | 0.366 | 0.004 | Rejected |
| H3: SV $\rightarrow$ INT | 0.170 | 2.622 | 0.009 | 0.051 | Accepted |
| H4: CV $\rightarrow$ INT | 0.292 | 5.203 | 0.000 | 0.188 | Accepted |
| H5: EP $\rightarrow$ INT | 0.031 | 0.667 | 0.499 | 0.003 | Rejected |
| H6: EV $\rightarrow$ INT | 0.216 | 4.021 | 0.000 | 0.110 | Accepted |

Note: CV = condition value; EV = emotional value; EP = epistemic value; ESI = ethical self-identity; FV (P) = functional value (price); FV (Q) = functional value (quality); INT = intention to adopt green vehicles; SV = social value.

### 4.4. Mediating Influence of Ethical Self-Identity in the Link between Consumption Values and the Adoption of Green Vehicles

Our conceptual framework postulated that ethical self-identity has a mediating influence on the relationship between consumption values and the intention to adopt green vehicles (Table 5). The findings reveal that ethical self-identity has a significant mediating influence in the relationship between functional value (quality) and the intention to adopt green vehicles ($\beta$ = 0.139, $p \leq$ 0.01), the relation between social value and the intention to adopt green vehicles ($\beta$ = 0.068, $p \leq$ 0.01), and the relationship between emotional value and the intention to adopt green vehicles ($\beta$ = 0.043, $p \leq$ 0.01). Collectively, these results support hypotheses H6 (a), H6(c), and H7(f). However, ethical self-identity does not positively mediate the relationship between functional value (price) and the intention to adopt green vehicles ($\beta$ = 0.001, $p \geq$ 0.05), the relationship between conditional value and the intention to adopt green vehicles ($\beta$ = 0.019, $p \geq$ 0.05), and the relationship between epistemic value and the intention to adopt green vehicles ($\beta$ = $-0.002$, $p \geq$ 0.05). Therefore, hypotheses H6(b), H6(d), and H6(e) are not supported.

**Table 5.** Mediating effect.

| Relationships | Beta Value | *t*-Value | *p* Values | Confidence Interval | Results |
|---|---|---|---|---|---|
| H6(a): FV (Q) → ESI → INT | 0.139 | 3.433 | 0.001 | [0.063–0.218] | Accepted |
| H6(b): FV(P) → ESI → INT | 0.001 | 0.073 | 0.942 | [−0.020–0.027] | Rejected |
| H6(c): SV → ESI → INT | 0.068 | 2.098 | 0.036 | [0.017–0.140] | Accepted |
| H6(d): CV → ESI → INT | 0.019 | 1.197 | 0.231 | [−0.012–0.050] | Rejected |
| H6(e): EP → ESI → INT | −0.002 | 0.106 | 0.915 | [−0.030–0.031] | Rejected |
| H7(f): EV → ESI → INT | 0.043 | 2.358 | 0.018 | [0.013–0.082] | Accepted |

Note: CV = condition value; EV = emotional value; EP = epistemic value; ESI = ethical self-identity; FV (P) = functional value (price); FV (Q) = functional value (quality); INT = intention to adopt green vehicles; SV = social value.

## 5. Discussion

This study offers novel understandings of green vehicle acceptance and confirms previous findings in the literature. The main objectives of this study were to investigate how intentions to adopt green vehicles among Generation Z are determined, through consumption values, and how ethical self-identity mediates the structural relationship between consumption values and the intention to adopt green vehicles.

Firstly, by examining the direct influence of consumption values on consumers' intention to adopt green vehicles, we discovered that functional value (quality) significantly influences the consumers' intention to adopt green vehicles; in this sense, the findings support past studies [65,89]. The results suggest that functional benefits, such as reliability, ride comfort, usability, range, and charging time, influence the intention to adopt green vehicles among Generation Z. Another reason could be the effective and effectual performance of green vehicles in reducing pollution; past studies mentioned that the perceived quality and performance of green products encouraged Pakistani consumers to participate in green consumption [90]. Our study found that price does not significantly influence consumers' intentions to adopt green vehicles among Generation Z; however, these findings are inconsistent with previous studies [58]. Our results suggest that Generation Z consumers in Pakistan are enthusiastic about paying a higher price for environmentally friendly vehicles to avoid the costs incurred by using conventional vehicles in the form of oil prices.

Our study found that social value significantly influences the intention to adopt green vehicles among Generation Z, a finding which contradicts some studies [58,65], although this is in line with others [2]. The reason could be that Pakistan is a collective society [63]. Consumers' purchasing decisions could easily be influenced by the opinions and suggestions of other people who are important to them, such as family, and, particularly, friends. This suggests that Generation Z consumers experience a need for social approval or enhancement to their social image when they adopt green vehicles. Our study found that conditional value significantly influences the intention to adopt green vehicles among Generation Z; these findings support past studies [69,71,82]. This indicates that discounts, promotions, and government subsidies are the main factors motivating the intention of Generation Z consumers to adopt green vehicles. In addition, governments need to take specific actions, such as providing subsidies to related manufacturers and importers and creating more opportunities to enhance conditional value. Our findings showed that epistemic value does not influence the intention to adopt green vehicles among Generation Z. These results contradict past studies [59,82,91]. Past studies have emphasized that Generation Z is more inclined toward social media platforms than other generational groups [92]. The possible reason for the non-significant relationship might be that Generation Z are aware of the green vehicles in the market and know about green vehicles. Therefore, they do not find any novelty that enhances their intention to adopt green vehicles. Our study found that emotional value positively impacts the intention to adopt green vehicles among Generation Z, supporting past studies [59,71,82]. This means that buying an environmentally friendly car, an act that is seemingly more conducive to preserving the environment, might, in turn, generate positive emotions among Generation Z; the feeling of doing the right thing for the environment and themselves. In addition, environmentally conscious purchasing

behavior is compatible with the protection of the natural environment. Therefore, it usually generates positive feelings.

Finally, we tested whether ethical self-identity mediates the relationship between consumption values and the intention to adopt green vehicles. To the best of our knowledge, this study is the first to investigate the impact of ethical self-identity on consumption values and the intention to adopt green vehicles. Our study found that ethical self-identity successfully mediates the relationship between consumption values (functional value (quality), social, and emotional value) and the intention to adopt green vehicles. This suggests that the specific attributes of green vehicles, such as environmental benefits, riding comfort, convenience of use or operability, reliability, and charging time influence ethical self-identity, leading consumers to adopt green vehicles. Similarly, our study also posits that Generation Z decides according to social influence rather than personal belief (in the case of green vehicles), which influences ethical-self-identity among this generation, who feel that they are doing the morally right thing for the environment and for society by adopting green vehicles. In addition, our study found that green vehicles evoke feelings and emotions in Generation Z, which ultimately enhances their ethical self-identity and encourages a sense of making a morally correct decision by adopting green vehicles.

Conversely, we could not find a mediating influence of ethical self-identity on the relationship between certain specific consumption values (functional value (price), epistemic value, and conditional value) and the intention to adopt green vehicles. This suggests that the economic benefit associated with green vehicles does not influence ethical self-identity among Generation Z. Similarly, our study found that conditional benefits, such as promotions, governmental subsidies, and rebates do not affect the ethical self-identity of Generation Z that ultimately influences them toward the adoption of green vehicles. In addition, our study also found that ethical self-identity failed to mediate the relationship between epistemic value and the intention to adopt green vehicles. The possible reason for this insignificant relationship could be that these consumers are familiar with the green vehicles in the market and have knowledge of green vehicles. Hence, the knowledge does not influence consumers to buy green vehicles. The likelihood that consumers will buy green vehicles, with or without this knowledge, is the same. As mentioned earlier, Generation Z consumers use social media to educate themselves before purchasing. Those who physically go to the showrooms with green vehicles are the ones who want advice, to learn more about a green car to make the best choice when buying it. Another reason is a shortage of charging stations; there is still a perception that there are not enough charging stations to keep green vehicles on the road.

### 5.1. Conclusion and Theoretical Contributions

Our findings have significant theoretical and practical implications. From an academic point of view, this study extends the current knowledge by incorporating the theory of consumption values and an ethical self-identity approach to studying consumer adoption behavior toward green vehicles. First, our results reveal that an emphasis on the functional value (quality), social value, and emotional value of green vehicles positively influences individual ethical self-identity and the intention to adopt green vehicles. However, the results also reveal that consumers are less anxious about the economic benefit and epistemic value gained from the adoption of green vehicles. Previous studies (such as, e.g., Qasim et al. [26], Pelsmacker et al. [73], and Lee et al. [56]) suggest that self-identities mediate the structural link between consumer motivation and sustainable behavior. In addition to these studies, we found that ethical self-identity affects the relationship between consumption values and the intention to adopt green vehicles.

Second, more recent research [93] shows that there is only minimal literature from a societal perspective on the public acceptance of green vehicles, especially among Generation Z, which leads to improvements in relation to the acceptance of green vehicles. Although its members are significantly affected by climate change, Generation Z was ignored in studies on environmentally friendly behavioral research, both at local and international

levels. This study is unique since it incorporates Generation Z, a segment of the population that remains largely under-investigated in sustainability research, despite its increasing importance and potential [91].

Thirdly, our results suggest that effective marketing campaigns can be achieved with the ethical identity of consumers in mind. Therefore, advertising should present products to consumers and enable them to recognize their role in taking moral action. In addition, our study found that conditional value positively impacts the intention to adopt green vehicles. The results also show that government subsidies and manufacturer promotions or discounts could generate positive attitudes among Pakistani consumers and encourage them to adopt green vehicles. Therefore, the government should consider pursuing tax exemption policies or other measures to boost green car sales to meet the green targets. This could make Pakistan the regional hub for energy-efficient vehicles.

Additionally, local manufacturers should launch their green vehicles to the market as soon as possible. Indeed, the green vehicle market is still at the penetration level, with a low proportion of electric cars being used in Pakistan. Similarly, we also find that functional value (quality), social value, and emotional value significantly influence consumers' intentions to adopt green vehicles. Therefore, it is recommended that marketers and practitioners promote environmentally friendly vehicles with marketing campaigns that arouse the social conscience of consumers as well as emotions regarding the environment and society. Finally, consumer concerns about quality should be highlighted. Since our study outcomes reveal that consumers are quality-conscious about green vehicles, manufacturers should ensure the quality of green vehicles in order to provide higher functional value for green vehicles. In addition, the positive attributes of green vehicles, such as environmental friendliness, safety, and comfort, must be guaranteed throughout the manufacturing process.

### 5.2. Practical Contributions

As already partially alluded to in Section 5.1, this study has significant implications for managers and decision-makers. Given that the results of this study emphasize the importance of consumer values and ethical self-identity, which will eventually contribute to the adoption of green vehicles, managers (e.g., the manufacturers of automobiles) should center their marketing campaigns on acknowledging and celebrating Generation Z's green values and ethical self-identity. Public policies aimed at spurring the adoption of green vehicles could utilize similar orientations by nurturing ecological values and self-identity. For example, financial incentives and fiscal measures could encourage the replacement of conventional cars with green alternatives by emphasizing that this replacement will better meet the core values and real identity of Generation Z customers (e.g., "unleash the real you"). The systematic association between green alternatives and Generation Z values and identity will foster Generation Z's natural affiliation with green alternatives. While the association between that generational cohort and green vehicles will be strengthened, this association might further apply to other products and categories.

### 5.3. Limitations and Future Direction

This study is focused on the intention to adopt green vehicles. Future research can be conducted on actual behavior, and comparisons of intentions and actual behavior can also be made to better understand how intentions turn into actual adoption behavior. Likewise, this study focused on a particular generation, Generation Z, in Pakistan. With caution, this means the study results can be generalized or practically applied to other generational groups, such as Generation Y (from Pakistan). Finally, the study is restricted in scope as it merely concentrated on consumption values and did not examine the consequences of any barriers (e.g., limited charging points and insufficient marketing). Since barriers are contemporary, and trade-offs may occur between barriers and values when purchasing decisions are made, the integration of barriers in future research will theoretically provide a better understanding of the motives behind the purchase of green vehicles.

**Author Contributions:** M.Y.B. and M.E. contributed to the conceptualization, formal analysis, investigation, methodology, writing of the original draft. M.A.K. and H.S. writing review and editing. All the authors contributed to the formal analysis, investigation, methodology, and writing review and editing. All authors have read and agreed to the published version of the manuscript.

**Funding:** This research received no external funding.

**Institutional Review Board Statement:** The study did not require a separate approval by the Ethics Board but it does follow ethics guidelines of authors' institutions. Additionally, authors had approval letter from their institution office to collect the data through survey from respondents.

**Informed Consent Statement:** Informed consent was obtained from all subjects involved in the study.

**Data Availability Statement:** The datasets analyzed during the current study are available from the corresponding author on reasonable request.

**Conflicts of Interest:** The authors declare no conflict of interest.

## Appendix A

**Table A1.** Measurment Items.

| | Construct/Items Description | Source |
|---|---|---|
| | *Functional Value (Quality)* | |
| FV(Q)1 | Green vehicles have a consistent quality. | Zailani et al. (2019) |
| FV(Q)2 | Green vehicles have an acceptable standard of quality. | |
| FV(Q)3 | Green vehicles are well made. | |
| | *Functional Value (Price)* | |
| FV(P)1 | Green vehicles offer value for money. | Qasim et al. (2019) |
| FV(P)2 | Green vehicles are reasonably priced. | |
| | *Social Value* | |
| SV1 | Buying a green vehicle would help me to feel morally acceptable. | Lin and Huang (2012) |
| SV1 | Buying a green vehicle product would improve the way that I am perceived. | |
| SV1 | Buying a green vehicle would make a good impression on others. | |
| | *Epistemic Value* | |
| EV1 | Before buying a green vehicle, I would obtain a substantial amount of information about different makes and models. | Zailani et al. (2019) |
| EV2 | I am willing to seek out novel information. | |
| EV3 | I like to search for what is new and different. | |
| | *Conditional Value* | |
| CV1 | I would buy a green vehicle instead of its conventional counterpart under worsening environmental conditions. | Zailani et al. (2019) |
| CV2 | I would buy a green vehicle instead of its conventional counterpart if there were a subsidy for it. | |
| CV3 | I would buy a green vehicle instead of its conventional counterpart if there were discount rates or promotional activity. | |
| | *Emotional Value* | |
| EV1 | For me, electric vehicles are the ones that I would enjoy. | Han et al. (2017) |
| EV2 | For me, the experience of driving electric vehicles would give me pleasure. | |
| EV3 | For me, an electric vehicle is the one that I would feel relaxed about using. | |
| | *Ethical Self-Identity* | |
| ESI1 | Ethics are important to me when making buying decisions. | Brich et al. (2018) |
| ESI2 | I think of myself as someone concerned about ethical issues. | |
| ESI3 | I think of myself as an ethical consumer. | |

**Table A1.** *Cont.*

|  | Construct/Items Description | Source |
| --- | --- | --- |
| | *Adoption Intention* | |
| INT1 | I would buy a green vehicle if the performance is the same as the conventional vehicles. | Mamun et al. (2019) |
| INT2 | I would buy a green vehicle even if it has a less appealing design. | |
| INT3 | When I replace my existing vehicle, I plan to purchase a green vehicle. | |
| INT4 | I intend to purchase a green vehicle next time because of its positive environmental contribution. | |

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
