# Peer review of "Investigating the Role of Ethical Self-Identity and Its Effect on Consumption Values and Intentions to Adopt Green Vehicles among Generation Z"

_sustainability, doi:10.3390/su14053015_

Round 1

Reviewer 1 Report

Dear Authors, 

The paper that you have submitted for evaluation is an interesting one and its topic is orginal but also a current one in light of the new found interest into green vehicles. At the same time after a careful examination of the paper, I recommend that the following adjusments be made: 

  • minor spell check  is required, there are places where you missed a letter or there is no space between words and references (see lines 83, 85, 126 etc.)#
  • at line 58 you mention ”the literature recommends”, which literature? is it the previous references? It is not clear, please reformulate
  • at line 62 - you use Pakistan twice in the same sentence, it does not sound ok. You could change the second with ”it”
  • line 65 you mention ”latest brodcasting” - please reference it
  • line 66 you mention ”vehicle fleet”, is it public or private vehicle fleet, or both? 
  • line 68 - 69 - the sentence starting with Therefore ...., doesn't make sense, it does not end well
  • line 71 - eliminate the word "less". The word "obliged" means forced to do something. Is not okay in the context of the sentence. 
  • line 99 - you mention Gen Y, please mention this cohort years, just like you did with Gen Z.
  • I recommend you ad a last paragraf for the introduction (at the end of it) with a brief description of the paper. 
  • line 132 - mentions "literature", please reference which one
  • lines 134 - 136, you have a sentence starting with "Consumer behaviors ....," it is not clear, I recommend you rephraze it
  • Also in lines 134 and 135 you mention the models of Ajzen and Fishbein, I recommend you take a look at the Technology Acceptance Models of Davis (1989) and Venkatesh and Davis (2000) https://www.researchgate.net/publication/227447282_A_Theoretical_Extension_of_the_Technology_Acceptance_Model_Four_Longitudinal_Field_Studies, it might bring more light in your theoretical review because we deal with green vehicles, which constitute technological items. 
  • After line 141 you present figure 1, but it does not have a linke to the text. Please develop several lines that link the theoretical framework with figure 1.
  • Line 155 - please add a source for Figure 1
  • Line 162 - you mention "quality and price" and later you use Q and P. in figure, in variable description etc. Please mention here that Quality is Q and Price is P. 
  • Line 170 - you miss the preposition "the" before authors
  • line 216 - you mention "new product or service" do you mean the usage of the product for the first time ever? If so, please clarify
  • line 280 - you mention 350 surveys. From what I know of research methodology, a survey is a research method, that uses the questionaire as a tool. Did you do 350 surveys? Or was it 350 questionaires, like you mentioned in line 288
  • at line 301 you mention that the items use din the questionaire are in the appendix. The final version of the paper should also include in the appendix the questionaire in itself, not just the variables
  • Regarding the methodology - please mention the names of the 2 universities, please underline the period of time that it took you to collect the responses. Please could you further explain the sampeling method. 
  • In line 303 you use the PLS-SEM abreviation first and in line 310 you use the full name of the analysis method. Should it not be the other way round? 
  • in line 484 you mention Conclusions and Theoretical Contributions, but you also provide some practical directions. I recommend that the section title reflect that. 

Author Response

We appreciate all the suggestions and comments given on our manuscript. Indeed, our work has been improved through these comments of the reviewers. The authors have revised and addressed all the concerns pointed out in different manuscript sections. 

Reviewer 2 Report

The topic of this paper is quite interesting. The theoretical framework of this paper looks fine. A lot of parts are well explained. However, all of the variables in the conceptual model have been widely examined in the existing literature. I really cannot see anything new from this paper. This paper cannot contribute new knowledge to the existing literature.

Author Response

We appreciate all the suggestions and comments given on our manuscript. Indeed, our work has been improved through these comments of the reviewers. 

Reviewer 3 Report

The publication of the paper is justified because the topic is treated in an original and appropriate way thanks to the development of issues related to ethical self-identity, consumption values and green vehicles through an interesting empirical research. Literature is developed in a clear way, but could be further expanded citing the main works about the themes treated. The methodology of the empirical research is developed and explained in a  understandable way also for a wide public responding appropriately with the paper objectives. Good presentation of the results, because the paper presents an important coherence between the objectives of introduction and methodology and the discussion. Also the conclusions are understandable but could be more expanded in response to the objectives set by the paper. The study is clearly explained with a good english level and an understandable technical language.  For greater deepening on the methodology: - Scarpato, D., Civero, G., Rusciano, V., & Risitano, M. (2020). Sustainable strategies and corporate social responsibility in the Italian fisheries companies. Corporate Social Responsibility and Environmental Management27(6), 2983-2990.    

Author Response

We appreciate all the suggestions and comments given on our manuscript. Indeed, our work has been improved through these comments of the reviewers. The authors have revised and addressed all the concerns pointed out in different sections of the manuscript.

Reviewer 4 Report

In this study, the authors integrate consumption values ​​theory and ethical self-identity to explore the intention to adopt green vehicles among generation Z.

The authors are focused on studying the values ​​of generation Z, but it is necessary to indicate how these values ​​differ from previous generations. In the Introduction, a slight digression into the ideological attitudes of generation Z and previous generations is necessary.
The authors need to remove the extra number: Line 157, "22.2 Hypothesis Development".
The section "Hypothesis Development" is placed in "2. Literature Review", which is quite reasonable from the standpoint of substantiating hypotheses. In such a case, the hypotheses should appear in the Introduction, along with a demonstration of the logic of dissection. In the next section, what was done to prove the hypotheses.
The theoretical substantiation also needs to demonstrate a critical attitude to existing sources in order to identify the uniqueness of one's research. Given that this study is fascinating refining its demonstration will allow the article to better communicate with the reader.
The "Discussion" section plays the role of Conclusions too. However, the Conclusion allows us to demonstrate the results of the study. The Conclusion will increase the comprehensibility of the text and the author's intention by readers. 

Author Response

(The authors gave the same response as above.)

Round 2

Reviewer 2 Report

I really appreciate the author(s) for their great effort into revising this paper based on previous comments. It looks much better than before. To further improve the quality of this paper, I propose my comments below.

  • In the introduction section, is it possible to change the research questions into the research objectives of this paper?
  • Is the pre-test conducted prior to the survey distribution?
  • Are there any problems with common method bias?
  • In the discussion section, please try to explain the reasons based on the insignificant hypotheses.

Author Response

(The authors gave the same response as above.)

Round 3

Reviewer 2 Report

I do believe that the author(s) have made a lot of effort into revising this paper based on my previous comments. However, one more comment is still not satisfied.

* If there are no problems with common method bias (CMB), I would recommend that the author(s) refer to the latest articles to see how to describe no CMB problems in this paper.

Author Response

Thanks for your precious comments, it really helps us a lot to improve the quality of my manuscript. CMB has been incorporated in the manuscript.